# Developing a Model of Risk Factors of Injury in Track and Field Athletes

**Zofia Wroblewska [1], Jacek Stodolka [2] and Krzysztof Mackala [2],***

[1] Faculty of Pure and Applied Mathematics, Wroclaw University of Science and Technology, ul. Janiszewskiego 14a, 50-372 Wrocław, Poland; Zofia.wroblewska12@gmail.com

[2] Department of Track and Field, University School of Physical Education, Wroclaw, Ul. Paderewskiego 35, 51-612 Wrocław, Poland; jacek.stodolka@awf.wroc.pl

* Correspondence: krzysztof.mackala@awf.wroc.pl; Tel.: +48-347-3147

**Abstract:** This work aimed to develop a model to assess the likelihood of injury in track and field athletes, and to establish which factors have the greatest impact. Tests verifying their significance were also reviewed, as well as the method for selecting variables. The key element was to confirm the quality of the classification system and to test the impact of individual factors on the likelihood of injury. The survey was carried out among physically active participants who take part in track and field sporting disciplines. The Cronbach's alpha was 0.73, which can be considered an acceptable value for the survey. The seven most important factors influencing the risk of injury were selected from a group of twenty-four and were used to create the model. The Nagelkerke's R2 reached 0.630 for the logit model, which indicates a good effect of the independent variables. The data suggested that the largest factor influencing the risk of injury was the number of prior injuries.

**Keywords:** risk factors; injury risk; survey data; athletics; training; health; modeling

## 1. Introduction

According to sociologists, the popularity of sport in the world is rising, reflecting the needs of a modern society seeking adrenaline rushes and high emotions [1]. The majority, however, are satisfied with watching sport or treating physical activity as a hobby [2]. Sport, for the remaining group, reflects their way of living [3]. Their ultimate objective is to find the limitations of their bodies and to achieve the best results [4]. In this case, the health benefits of sport become dangerous. Where there is regular physical activity, especially in running, there is always a risk of injury. According to Wiese-Bjornstal [5] and Fernandes et al. [6], the main causes of sport injuries are physical, biological including physiological, anatomical, and some training factors, such as muscle imbalances, overtraining, physical fatigue, or a lack of base physical condition. Environmental factors, such as facilities and unsuitable sport equipment, can also affect the occurrence of injury. Additionally, injuries often cause severe stress for athletes, disrupt their training, interfere with high performance competition, and can lead to feelings of separation and isolation from their family, coaches, and teammates [6,7].

Injuries related to sport, cars [8], and bikes [9], or accidents in the workplace, are a daily occurrence in human lives. The usual outcome is some kind of injury that takes place during competitive sport as well as during recreation activities, where people should have fun and rest. Bare and Holne [10] claimed that according to the van Mechelen [11] model, once it has been recognized through injury investigation that sports injuries create a threat to the health of athletes, injury prevention must be established. This includes information on why a particular athlete may be at risk in a given situation (risk factors) or how injuries happen [9,10]. A daily training regime and, consequently, excessive exercise frequently results in injuries. It is challenging to predict specific injuries since the entire process is very complex

and depends on many factors [12]. The risk of injury can, however, be assessed, and the most critical factors with the most significant influence can be determined. Injury modeling in sports was the objective of this study. According to Alizadeh et al. [13], considering injury risk factors is essential to identify injury-prone athletes and to develop injury-prevention strategies. As injury causation is usually complex, risk factors must be known by earlier intervention. There is extensive literature on the subject. Numerous studies have evaluated sports injury safety management in different types of sport [14,15]. These studies have assessed several factors such as type of sport, gender, age, type of movement structure, and fatalities. For example, Hopkins [16] reviewed measures useful in analyzing the risk of injuries. Bahr and Holme [10] determined injury risk factors. In both types of research, the authors focused more on methods, which are quite similar to those considered in this work, but not on real data. The combination of established methodology and analysis of real data forms the novelty of this paper.

The occurrence of injury is a result of multiple factors, which means that their classification is difficult. Some researchers [17] have analyzed injury proportions among runners. They divided the injuries into two important rating divisions: incidence proportions and prevalence rates. It seems that this is an insufficient classification, due to a large discrepancy in the proportion of injuries found between researchers. In athletics, most of the research on the occurrence of injury has been carried out on runners, including sprinters. There is little data regarding injuries in athletes of jumping and throwing events. Despite the health benefits of running [18], injuries are very common, with incidence rates ranging between 18.2% and 92.4% [19–21]. In turn, Kluitenberg [17] claimed that a large differences in time-loss injury proportions exist among various running specialties; 3.2% in cross-country runners to an injury proportion of 84.9% in beginners. Another study [22] stated that about 25% of runners are injured at any given time, and about half experience an injury that takes them out of running for a period during any year. According to van der Worp [23], higher-quality studies of risk factors for running injuries are required before conclusions can be made. This is due to the very high risk of getting injured in running activities. This high injury rate, he concluded, is between 19% and 79%, and is mainly due to differences in the definition of injury, study populations, and follow-up periods [24].

However, there are few studies that evaluate the risk of injury based on mathematical models concerning track and field events. Some models try to establish a connection between psychological background and the occurrence of sports injuries [13]. One of the most well-known models is Williams and Andersen's [25], stress-injury model. The model suggests the effects of psychological risk factors on injuries and other sport-related health results. Another model is Junge's model of the effect of psychological factors on sports injury [26], with three distinct psychological categories counting: coping resources, psychological stressors, and emotional state [27]. Another one is Johnson and Ivarsson's [27] empirical model of injury risk factors. This model highlights personality factors and stress.

This work aimed to develop a model to assess the likelihood of injury in track and field athletes, and to establish which factors have the greatest influence on the probability. The emphasis was placed on the analysis of the most frequently occurring injuries in athletics that are not classified as having extensive consequences for the athlete's health, but require the help of specialists, as well as significant changes in the athlete's training program [28].

## 2. Material and Methods

### 2.1. Study Design

This study is focused on identifying and determining the cause of injury occurrence among track and field athletes of different events, different age categories, and performance level. The necessary information was obtained using a specially constructed questionnaire, including information on why a particular athlete may be at risk in a given situation or how injuries happen. Cronbach's alpha was used to assess the validity of the questionnaire. The next step was selecting the risk factors that influence

expected injuries and establishing relations between them. Logistic regression analysis was used to analyze the data obtained. Once it has been recognized that sports injuries threaten the health and performance of athletes, the reasons must be established for future injury prevention. Here, a theoretical approach was used. Modeling the risk of injury in sport assesses the most common injuries and their causes. The probability of injury was modeled using binomial and link distribution and a logit function was established. The Hosmer–Lemeshow (HL) test was applied and WoE coefficients (weight of evidence) for all model variables were calculated. The design and implementation of this model may be used in the future for modeling the risk of key injuries among runners, which probably constitute the majority of injuries among track and field athletes.

### 2.2. Subject

For this study, survey data and identifying factors were collected from a group of physically active track and field athletes, including jumpers, runners, and throwers. The sample of respondents included five age categories: U16, U18, U20, U23, and seniors (athletes between 24 and 35 years old). Moreover, the athletes were at different levels of sports performance: international, national, first, second, and third sports classes. The sample of respondents included 105 females and 101 males (Figure 1), which represents 5.5% of the total number (3750; 1938 women and 1812 men) of competitors registered in 2016–2017 by the Polish Athletics Association. I hereby confirm that I have taken into account all ethics issues. The research does not involve physical interventions on the study participants (questioner) and did not, at any stage, involve animals. Therefore, the study (W13_221661_2018) was accepted by the Human Ethics Committee of the Wroclaw University of Science and Technology, and does not require an ethics code.

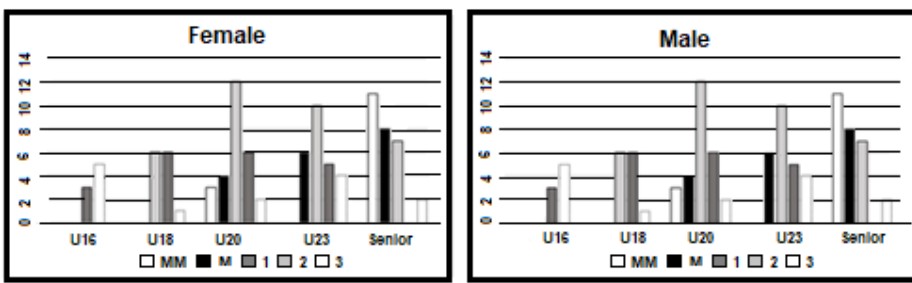

**Figure 1.** Quantitative characteristic of female and male athletes participating in the experiment, divided into sports classes (level of performance).

### 2.3. Injury History Survey

The first step toward injury evaluation and future modeling was selecting the factors that influence expected injuries and establishing relations between them. Different tools were used to acquire data and select these factors [29]. For this study, survey data were prepared and analyzed. The questionnaire was part of the first author's dissertation [30]. In our experiment, the reportable concept of injury should be understood as the physical condition of the athlete, who, at least once in one year (between August 2016 and August 2017) of the training period, was temporarily (one to two months) prevented from continuing training or participating in competitions. This injury required medical advice and short rehabilitation. The injury is recognized as the specific type of injury (characteristic of athletics) characterized by a similar recovery time. They concerned the following health problems: muscle pulling (hamstring, quadriceps, groin, and other), Achilles tendonitis, knee tendonitis, ankle sprain, wrist twist, or lower back pain. The injury specification applies to all athletes irrespective of sport class, age, gender, and event. The electronic survey data was collected from athletes in October 2017, after finishing the 2017 competition period.

The questionnaire was divided into three sections/categories concerning demographic factors, training/competition factors, and health/regeneration factors. Demographics included sex, BMI, and

morphology. The training/competition section contained information about training specification, including training load, warm-up connected with a single training session, training experience, sport level availability and quality of sport facility, atmospheric condition in summer training period and competition period, and biomechanics, which is connected to the techniques executed during training and competition. They were categorized in such a way that the outliers (values three standard deviations away from the mean) were included in one data set. More frequently occurring answers were further classified into sets. Adjacent categories for ordinal scale variables were combined when the number of observations in a selected category was not higher than five (Table 1). The majority of the factors were categorical variables for which coding is an important process. The purpose of coding is to prepare data for multidimensional analysis, which has a great influence on the interpretation of coefficients in models. All variables are coded as 0 or 1. A variable taking the value 0 is called a reference category. Dummy coding is used in multidimensional data models and is designed to answer the query of how an estimated coefficient variable $\hat{\beta}_j$ of $j$ in each analyzed category differs from the results of this coefficient for the reference category.

**Table 1.** Names of categorical predictors divided into numerical ranking (categories and their amount).

| Section/Predictors | Numerical Ranking | Section/Predictors | Numerical Ranking | Section/Predictors | Numerical Ranking |
|---|---|---|---|---|---|
| Demographic | | Training/Competition | | Health/Regeneration | |
| | | | | previous injuries | 4 |
| | | | | resistance to illness | 4 |
| | | athletics competition | 4 | wellness | 4 |
| | | sports level | 5 | mental load | 2 |
| sex | 2 | training load | 5 | natural regeneration | 4 |
| morphological | 2 | warm-up | 4 | ability | 5 |
| BMI | 5 | sport facility | 5 | sleep | 3 |
| | | atmospheric conditions | 4 | physiotherapy | 2 |
| | | technique | 2 | diet | 2 |
| | | | | supplementation | 3 |

The logical supplement to the categorical division within which there is a likelihood of injury in the mobility aspect, characteristic for athletics, is the introduction of numerical ranking, from 1 to 5, of selected factors/observations. The numerical ranking was performed as follows:

- BMI rate: 1: ≤16; 2: 16–18; 3: 18–20; 4: 20–22; 5: >22.
- No. of competition: 1: 0; 2: (1–3); 3: (3–10); 4: (10–20); 5: >20.
- Sports performance level: 1: International (MM); 2: National (M); 3: First class (1); 4: Second class (2); 5: Third class (3).
- No. of previous injuries: 1: 0; 2: 1; 3: 2 or 3; 4: >3.
- Frequency of training/competition in bad atmospheric conditions: 1: never; 2: occasional; 3: quite often; 4: often 5: always.
- Frequency of practicing/competition in a badly prepared area of a sports facility: 1: never; 2: occasional; 3: quite often; 4: often; 5: always.
- Quality of warm-up: 1: never; 2: always; 3: good; 4: very good,
- Training load: 1: no training load; 2: small; 3: medium; 4: heavy; 5: very heavy.
- Resistance (number) to illnesses: 1: never; 2: 0–2; 3: 3–5; 4: >5,
- Hydration: number of water liters per day 1: <1 L; 2: 1–2 L; 3: >2 L.
- Frequency of physiotherapy application per year: 1: 1–5; 2: 5–20; 3: 20–100; 4: >100.
- Frequency of health treatment: 1: 0–5; 2: 6–20; 3: 21–50, 4: >50.
- Sleep quality in no. of hours: 1: <5; 2: 5–7; 3: 7–8; 4: 8–9; 5: >9.

When the number means sex, 1: injury among women; 2: injury among men. The following factors are also included in the binary numerical ranking:

- Biomechanics: quality of posture: 1: correct; 2: with defects.
- Natural regeneration ability: 1: insufficient; 2: good.
- Supplementation: 1: insufficient; 2: supplementation.
- Diet: 1: insufficient; 2: diet.
- Mental load: stress caused by training and competitions 1: no load; 2: load.

### 2.4. Statistical Procedure

Logistic regression analysis was used to analyze the data obtained [31–33]. It aimed to determine the approximate relationship between a dependent variable (represented as $y$) and an independent variable (represented as $x$). An independent variable was referred to as a factor. If $p$ factors are obtained, our data for $n$ athletes can be represented inFigure 2.

$$\begin{array}{cccc} y_1 & X_{11} & \dots & X_{1n} \\ \dots & \dots & \dots & \dots \\ y_n & X_{n1} & \dots & X_{np} \end{array}$$

**Figure 2.** Logistic regression—the pattern of approximate relationship between a dependent variable (y) and an independent (x) variables.

The $x_{ij}$ is the value of factor $j$ for element $I$, while $y_i$ is a value of the dependent variable for this element. Each observation $y_1$, $y_2$, ..., $y_n$ has one of the two possible outcomes: 0 for no injury (failure), or 1 for an injury (pass). The following chapters describe them as predictors or outcomes. There were qualitative variables: nominal and ordinal, which have categories, or quantitative variables. To improve the survey, some quantitative variables were divided into categories despite losing a part of the information. Cronbach's alpha was used to assess the validity of questionnaire. Cronbach's alpha = 0.73, which is considered an acceptable value. Cronbach Alpha values range from 0 to 1. In most cases, the value should be at least 0.70 or higher, although a value from 0.60 to 0.70 is acceptable. For the calculation of effect size in logit models, the coefficient of Cox and Snell was used. This coefficient takes values between 0 and 1, where 0 indicates a very weak effect of the independent variable. However, this coefficient cannot reach a value of 1; therefore, Nagelkerke's R2 was applied, which is the value of Cox and Snell's R2 standardized on the maximum value it can achieve.

The Statistica 10.0 software divides the factors into uncategorized, quantitative predictors and categorized, categorical, and numerical predictors.

## 3. Results

Forward stepwise selection was used to choose seven predictors for the model: previous injuries, sleep, age, blood, training load, atmospheric conditions, and competition. Table 2 shows that according to S and Wald statistics, the other predictors should be removed from the model. This model will be referred to as M. Table 2 presents the last step of this method, confirming the choice of model M.

Factors included in model M were classified into three groups introduced in Table 1:

- age–demographics,
- sleep, blood, previous injuries, health/regeneration,
- training load, atmospheric conditions, training/competition.

In our studies, the HL test value was 7.36, which produced a $p$-value equal to 0.5. The $p$-value was higher than the standard significance level of $p = 0.05$. This means there was no evidence to reject the hypothesis that the analyzed model of logistic regression was well fitted to the data. This was also confirmed by the high *AUC* value of 0.88. Figure 3 presents the receiver operating characteristic (ROC) curve and area under the curve (AUC) value.

**Table 2.** Modeled probability of injury using binomial and link distribution and logit function.

| Step 9 | | Model Creation Distribution: BINOMAIL, LINK Function: LOGI TModeled Probability of Injure = 1 | | | | Model Creation |
|---|---|---|---|---|---|---|
| Effect | df | Wald Stat. | Wald p | S. pkt Stat. | S. pkt p | |
| Previous injuries | 3 | 40.66 | 0.00 | | | In model |
| Age | 4 | 18.30 | 0.00 | | | In model |
| Sleep | 1 | 10.27 | 0.00 | | | In model |
| Blood | 1 | 2.55 | 0.11 | | | In model |
| Competition | 3 | 4.71 | 0.19 | | | In model |
| Training Load | 4 | 9.67 | 0.04 | | | In model |
| At. cond | 3 | 9.08 | 0.04 | | | In model |
| Obj. cond | 4 | | | 0.27 | 0.99 | Out |
| Warm-up | 3 | | | 0.99 | 0.80 | Out |
| Experience | 1 | | | 0.42 | 0.51 | Out |
| BMI | 4 | | | 5.17 | 0.27 | Out |
| Level of performance | 4 | | | 2.21 | 0.57 | Out |
| Sex | 1 | | | 0.00 | 0.96 | Out |
| Diet | 1 | | | 2.21 | 0.14 | Out |
| Biomechanics | 1 | | | 0.45 | 0.50 | Out |
| Supplementation | 3 | | | 0.54 | 0.46 | Out |
| Regeneration | 2 | | | 0.78 | 0.85 | Out |
| Physiotherapy | 3 | | | 0.74 | 0.69 | Out |
| Wellness | 2 | | | 2.43 | 0.49 | Out |
| Illness | 1 | | | 1.92 | 0.16 | Out |
| Summer performance | 1 | | | 0.00 | 0.98 | Out |
| Winter performance | 1 | | | 0.59 | 0.44 | Out |
| Injuries | 4 | | | 6.50 | 0.16 | Out |
| Drinks | 2 | | | 1.76 | 0.44 | Out |

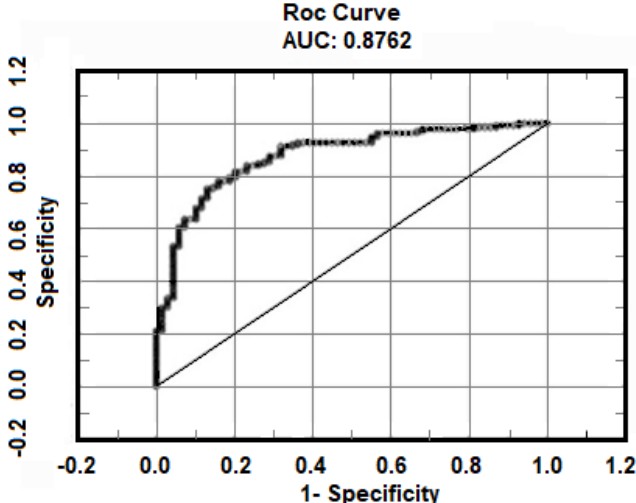

**Figure 3.** Receiver operating characteristic (ROC) curve for model M.

The odds ratio for model M calculated from the classification matrix (Table 3) was 19.53. It was much higher than one and indicated that the classification was much better than that expected "at random".

**Table 3.** The classification matrix for model M.

| | Case Classification Odd. Ratio 19.531250 Log Odd. Ratio2.972016 | | |
| --- | --- | --- | --- |
| | **Predicted: 1** | **Predicted: 0** | **Correct %** |
| Observed: 1 | 125 | 12 | 91.24 |
| Observed: 0 | 24 | 45 | 65.22 |

Next, WoE coefficients for all model M variables were calculated. The strongest relationship was observed between the factor *prv.inj* (number of previous injuries) and the risk of injury, reflected by the highest value of IV = 0.993.

Analysis of the graph presented in Figure 4 and WoE coefficients showed that the risk of injury increased with the category. The higher the category, the greater the number of previous injuries and the lower the WoE coefficient. The lower the WoE coefficient, the higher the risk of injury.

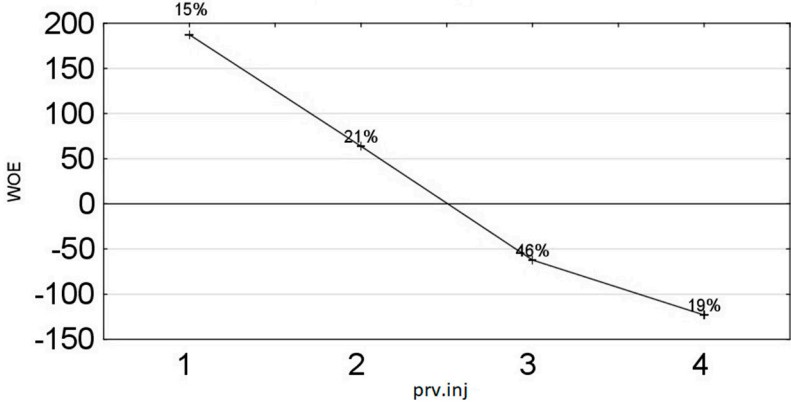

**Figure 4.** WoE (weight of evidence) graph for the variable *previous injuries*.

The WoE coefficient gradually decreased (Figure 4) across categories 1 and 2 (positive WoE, risk of injury is relatively small for elements included in these classes) to categories 3 and 4 (negative WoE, increase in the risk of injury).

The training period was the second predictor creating the strongest relationship with the risk of injury, for which the IV value was 0.326. The IV statistic of *age* was frequently related to the training period was only slightly lower (IV = 0.282). There were mostly moderate relationships between the remaining predictors and probability of belonging to the class 0.1 < IV < 0.3.

## 4. Discussion

This work aimed to develop a model to assess the likelihood of injury in track and field athletes, and to establish which factors determine the probability to the greatest extent. Our primary focus was on severe injuries with extensive consequences and require the help of specialists as well as significant changes in the training program [28]. There were also large discrepancies between the group of respondents (a) and the representative group (b) in females and males. This was due to the fact the questionnaire was filled mainly by athletes with higher sports classes.

According to van Mechelen [11], the injury risk factors are mostly divided into two main categories: internal—athlete-related risk factors, often recognized as intrinsic factors; and external—environmental risk factors. When we try to model the risk of injury in sport, all these factors can be divided into modifiable and nonmodifiable factors [10]. In our studies, most of the factors can be classified into the above categories; however, our analysis is based on three categories: demographic factors, training/competition factors, and health/regeneration factors. The more precise the division and

grouping of factors, the greater the likelihood of classifying a particular factor as modifiable. Such an assumption may indicate a more effective modeling process, and thus help to determine the most critical factors affecting sports injuries. Nonmodifiable risk factors, such as gender and age, may be of interest.

Because age and gender are nonmodifiable, they were excluded from the model. Most factors are training factors that are modifiable through behavioral approaches [10]. Therefore, such a multifactorial approach to the determination of risk factors of sports injuries requires a dynamic model, which takes the entire sequence of events preceding or directly affecting the occurrence of an injury. According to Meeuwisse [34], the sum of these risk factors and the interaction between them "prepares" the athlete for an injury to occur in a given situation. Studies considering only a single variable and its impact on injury risk may be too simplistic. Therefore, in order to better understand the etiology of injuries, the collective involvement of many factors in the risk of injury should be analyzed [35,36].

According to Ruddi [35] and Bittencourt [36], injuries occur as a result of complex and nonlinear interactions between multiple factors. Bahr [37] stated that it is unlikely that a single, isolated factor is capable of providing enough information to predict injuries at the individual level. Our model includes seven predictors: previous injuries, sleep, age, blood, training load, atmospheric conditions, and competition. Although age belongs to the unmodified category of risk factors [10], in our study, age was included in the modeling process because the athletes were divided into age categories. Training and sports level, as well as competitions, are related to age categories, including the number of competitions in the season. Training factors are modifiable through behavioral changes, which can alter blood parameters. Training and competition periods occur in an annual training cycle. The training period was the critical predictor creating the strongest relationship with the risk of injury, for which the IV value was 0.326.

Track and field athletes are at particularly high risk of injury due to their involvement in many events. This shows that the inherent risk of sports injury is related to the degree of athlete event exposure [38,39]. The more one trains and competes, the more one is exposed and the more injuries occur. The frequency and number of injuries as well as the significance of the injury influence further training and competition. Therefore, the most critical risk factors affecting sports injuries are health-related factors, such as previous injuries and sleep, which are associated with resistance to illness [40]. This was confirmed by our model M, which showed strong connections between the factor *prv.inj* (number of previous injuries) and the risk of injury, reflected by the highest value of IV = 0.993. A study done by Chen [40] showed that 71.1% of athletes who have suffered from sports injuries in the past were likely to suffer again. A study by Walter et al. [41] pointed out that only 50% of sports injuries are new, while the rest are repeat injuries. The explanation of these dependencies highlights Ruddy's [35] claims that previous injury had been used as an example to explain methodologies that can be used to determine the association between a factor and the risk of injury. These dependencies confirm Akobeng [42], who pointed out that all factors associated with an increase or decrease in the risk of injury are often repetitive.

Modeling the risk of injury in sport is a theoretical analysis of the risk of getting injured by paying attention to the most common injuries and what may cause them. The design and implementation of this model may be used for modeling the risk of car accidents [8] or accidents in the workplace.

We agree with Carey's [43] suggestion that a significant limitation in implementing complex approaches of modeling the risk of injury is the amount of data. It is required for the application of the appropriate methodology of investigation. Implementation of sufficient data will help determine the most critical interactions between risk factors and, above all, find those factors that are most often repeated and play a serious role in the occurrence of injury. Therefore more detailed injury data in future research is needed to reinforce the present results. The second limitation is the small number of athletes participating in the study. Further research should more deeply investigate the relationship between training load data, especially considering the training and competition cycle and the risk of injury. The limitations of the present survey-based study also include answers based on the subjective

feelings of the respondents. The final thoughts were based on factors that can be clearly identified by the respondents.

## 5. Conclusions

According to its AUC and IV values, previous injuries (*prv.inj*) was revealed to be the most significant factor. There is no doubt that each injury has a minor or major influence on the biomechanics of athletes' movement, causing unnatural loading and often leading to other injuries. Training experience (*experience*) was the second strongest predictor affecting the probability of injury. The risk of injury increases with each year of training. Although training and exercise prepare the body for loading and may protect it from injury, the final analysis showed that the potential risk of injury over time is increasing. The significance of these two factors is highly intuitive and commonly known among athletes, thus these findings support the correctness of the presented method. The Cronbach's alpha was 0.73, which can be considered acceptable. The Nagelkerke's R2 reached 0.630 for the logit model, which indicates a moderately strong effect of the independent variables. Our results indicate that early identification of risk factors and their gradation will help prevent further injuries.

**Author Contributions:** Conceptualization, Z.W., K.M.; Methodology, Z.W., K.M., J.S.; Software, Z.W.; validation, Z.W., J.S.; Formal analysis, Z.W., K.M., J.S.; Investigation, Z.W., K.M., J.S.; A resources, J.S.; Data curation, Z.W., K.M.; Writing—Original draft preparation, Z.W., K.M., J.S.; Writing—Review and editing, Z.W., K.M.; Visualization, Z.W., K.M.; Supervision, K.M., J.S. All authors have read and agreed to the published version of the manuscript.

**Funding:** This research received no external funding.

**Acknowledgments:** The authors would like to acknowledge the involvement of the participants for their contribution to this study.

**Conflicts of Interest:** The authors have no conflict of interest to declare. The results do not constitute an endorsement of any product or device. The authors would like to thank the sprinters who participated in this study.

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
