# Peer review of "Developing a Model of Risk Factors of Injury in Track and Field Athletes"

_applsci, doi:10.3390/app10082963_

Round 1

Reviewer 1 Report

Thanks for letting me read the paper. Although I have enjoyed reading it, I find some issues that need to be improved. I understand that once these issues are improved the work is likely to be published.
1. Attached is an overlay report indicating a moderate level (15.8%).
2. I consider that the use of the word model in the title is not appropriate. The statistical analyses carried out do not allow us to assume this situation. I understand that to create a model you must use System of Structural Equations and/or Mediation Analysis.
3. In the summary you should include the main results and the size of the effect.
4. They should improve the key words and adapt them to the substantive content of the work.
5. L.32-35: It would be important to provide recent statistics on the number of injuries per sport or per specialty.
6. L.49 et seq:I suggest that you review Vol 23, No 2 (2014) of the Journal of Sport Psychology.
7. L. 76: This section should be called Method.
8. L.77: This section should be called Participants. And before this section, it should include a section called "Research Design.
9. L.94: The survey should be included in an annex to understand the scope of the survey.
10. L.165: Care should be taken to ensure that there is no collinearity.
11. L.181: The normality of the sample should be studied.
12. Bibliographical references should be updated.

Reviewer 2 Report

I propose the authors to make a table with the descriptive data of participants: age, sex, categories, BMI etc. The figures are not relevant to present the participants.

The authors must correct why all participants must receive the parental consent, see lines 86-87. For participants over 18 years old must this consent??? this is understand in the text....please detailed this.

Please include the number of Ethical agreement, see line 88.

Also, please detailed who validated the questionnaire and which is the value of alpha?

The headers of tables must rearrange, the p-value must decimal.

Also, please include the DOI for each article present in References. 

Round 2

Reviewer 1 Report

I thank the authors for their efforts in answering the questions raised. However, there are two questions to be asked:
1. I do not agree with the authors in the answer they give about the size of the effect. For the calculation of effect size in logit models, the Pearson determination coefficient can be used, which are generically called: Pseudo-R2. The coefficient of Cox and Snell can also be used. This coefficient takes values between 0 and 1 so that 0 would indicate a very low effect of the independent variables, while in the vicinity of 1 it would show a considerable effect. However, this coefficient cannot reach the value of 1. That is why Nagelkerke's R2 is used, which is the value of Cox and Snell's R2 standardized on the maximum value it could take. This ensures that its value can be interpreted between 0 and 1. Please calculate the size of the effect and include it in the summary. This would allow this paper to be included in further meta-analysis research.
2. I understand the rules of the journal but they must include the research design where, in addition to indicating the type of design, it is justified and referenced.
Having solved these two questions, I consider the article ready for publication.
Thank you.

Reviewer 2 Report

The authors made the required changes.
